# Increased Likelihood of Falling in Older Cannabis Users vs. Non-Users

**DOI:** 10.3390/brainsci11020134

**Published:** 2021-01-21

**Authors:** Craig D. Workman, Alexandra C. Fietsam, Jacob Sosnoff, Thorsten Rudroff

**Affiliations:** 1Department of Health and Human Physiology, University of Iowa, Iowa City, IA 52242, USA; craig-workman@uiowa.edu (C.D.W.); alexandra-fietsam@uiowa.edu (A.C.F.); 2Department of Kinesiology and Community Health, University of Illinois at Urbana-Champaign, Champaign, IL 61820, USA; jsosnoff@illinois.edu; 3Department of Neurology, University of Iowa Hospitals and Clinics, Iowa City, IA 52242, USA

**Keywords:** cannabis, fall risk, gait, balance, older adults

## Abstract

Cannabis is one of the most common drugs in the United States and is the third most prevalent substance consumed by adults aged 50 years and older. Normal aging is associated with physiological changes that make older adults vulnerable to impaired function and geriatric conditions (e.g., falls, cognitive impairment). However, the impact of medical cannabis use on fall risk in older adults remains unexplored. The purpose of this study was to investigate if cannabis use in older adults influences fall risk, cognitive function, and motor function. It was hypothesized that older chronic cannabis users would perform worse than non-users on gait, balance, and cognitive tests. Sixteen older adults, split into cannabis Users and age- and sex-matched Non-Users groups (n = 8/group), participated in the study. The results indicate a higher fall risk, worse one leg standing balance performance, and slower gait speed in Users vs. Non-Users. No significant differences in cognitive function were found. Thus, chronic cannabis use was purported to exacerbate the poorer balance control and slower gait velocity associated with normal aging. Future mechanistic (e.g., neuroimaging) investigations of the short- and long-term effects of using a variety of cannabis products (e.g., THC/CBD ratios, routes of administration) on cognitive function, motor function, and fall incidence in older adults are suggested.

## 1. Introduction

Cannabis is one of the most common drugs in the United States and is the third most prevalent substance consumed by adults aged 50 years and older [1]. Han et al. [2] found a significant increase in the prevalence of past-year cannabis use in the United States among older adults from 2006/07 to 2012/13, with particularly large relative increases among those aged 65 and older. Given the high rates of substance use among the Baby Boomer generation, this increase in cannabis use was not unanticipated; furthermore, considering that the majority of Baby Boomers have still not yet reached 65 years of age, this trend is expected to continue into the next decade. Additionally, as a person ages, the chances of acquiring a disease or disability that is possibly amenable to medical cannabis, e.g., pain, cachexia, nausea, and other diagnosable conditions associated with cancer [3], increases considerably, further augmenting the likelihood of its use among older adults.

With the increased availability and use of cannabis by older adults, the benefits and risks of cannabis need to be as rigorously evaluated as with other substances. For example, alcohol use among older adults has been extensively studied, allowing for an adequate understanding of the specific risks [4,5,6]. This risk analysis has informed guidelines for limits of alcohol consumption for older adults [7], screening for its use, and treatment for its abuse [8]. However, little to no research has explored cannabis use among older adults in the same manner. While studies examining the benefits of cannabis for medicinal use are emerging, e.g., for chronic pain [9,10] and muscle spasticity [11], the risks for older adults are still unclear.

Typical aging is characterized by physiological changes that make older adults vulnerable to impaired function, chronic disease, and geriatric conditions such as falls and cognitive impairment [12]. Falls are causally linked with morbidity and mortality and are the primary source of injuries, both fatal and nonfatal, among older adults. According to a 2014 Centers for Disease Control and Prevention analysis of data from the Behavioral Risk Factor Surveillance System survey in the United States, approximately 28.7% of older adults disclosed at least one fall in the previous 12 months, which equated to ~29 million falls and ~7 million fall-related injuries [13]. Of the fallers, 37.5% reported that at least one of their falls required medical treatment or limited their movement [13] and ~27,000 deaths resulted from falls in this same period. The average Medicare expenditure is approximately USD 10,000 per fall and this cost doubles for individuals ≥ 72 years old. Thus, medical costs for falls in older adults are estimated to be USD 50 billion annually [14]. As the population ages, the number of falls, fall-related injuries, and costs of falls are also anticipated to increase substantially [15].

Falls and cognitive impairment are a “well-known couple” [16]; older adults with moderate to severe cognitive impairment have a higher fall risk, with an annual incidence of around 60–80%, which is twice the rate of cognitively normal older adults [17]. Importantly, impaired cognitive function is a known effect of cannabis use [18], and there is increasing evidence that those effects may persist later in life [19,20,21,22]. It is also known that cannabis has an impact on cognitive–motor skills and the brain mechanisms that modulate coordinated movement in regular cannabis users. These studies have shown reduced neural activity in frontal brain networks that were associated with an increased fear of falling in older adults [23,24]. For example, Pillay et al. [25] reported decreased activation in the supplementary motor area in regular cannabis users during their performance of a motor function test. Therefore, disruptions of these networks may pose a risk of falls through impairment of motor control.

In addition, the other physiological effects of chronic cannabis use (e.g., influences on CB1, which is broadly distributed in the brain and spinal cord [26,27]) may further increase fall risk. It is already well known that other medications prescribed to treat similar symptoms as medical cannabis use described above (e.g., opioids, antiepileptic medications, and polypharmacy) are significantly associated with increased fall risk. However, the impact of medical cannabis use on fall risk in older adults remains unexplored [28]. Therefore, the purpose of this study was to investigate the fall risk, cognitive function, and motor function of older individuals (> 50 years) who use cannabis for medical purposes. It was hypothesized that older chronic cannabis users would have a higher fall risk, poorer motor performance, and reduced cognitive performance than older non-users.

## 2. Methods

### 2.1. Subjects

Eight medical cannabis users (Users) and eight sex- and age-matched controls (Non-Users) were recruited (subject demographics are listed in Table 1). Inclusion criteria included (1) being between the ages of 50 and 80 years old, (2) healthy enough to complete the protocol based on information obtained from a clinical exam and past medical history, (3) part of the Iowa Medical Cannabidiol program (user group) or have not used cannabis in the last five years (Non-User group), (4) able to comprehend the protocol, as indicated by the ability to respond to questions about the study after reading the consent form, (5) able to use and be contacted by telephone, and (6) able to speak, read, and understand English and complete a questionnaire in English. Exclusion criteria included (1) pregnancy, (2) history of traumatic brain injury, and (3) other drug use or alchoholism. This study was approved by the Institutional Review Board at the University of Iowa (IRB#201909808, 17 December 2019), performed in accordance with the Declaration of Helsinki, and all subjects provided written consent before participating.

### 2.2. Experimental Protocol

Subjects completed one experimental session. After signing the consent form, a urine test (iScreen IS1THC dipstick; Alere Toxicology, Portsmouth, VA, USA) was conducted on all subjects to detect the presence of cannabis and to ensure group assignment (Users vs. Non-Users). Subjects then completed Item 14 of the Berg Balance Scale (BBS-14), question 1 of the Activities Balance Confidence (ABC-1) scale, the 9-Hole Peg Test (9-HPT), the Deary–Liewald reaction time (RT) tasks (simple and choice RT) [29], the Flanker Inhibitory Control and Attention Test, static posturography, and a 30-m walk test.

### 2.3. Measurements

#### 2.3.1. Fall Risk

Fall risk was assessed using a model developed by Lajoie and Gallagher [30], who showed that BBS-14, ABC-1, and simple RT were associated with fall status with 91% sensitivity and 97% specificity. BBS-14 asks subjects to stand on one leg for at least 10 s [31] and ABC-1 asks subjects to rate how confident they are that they will not fall when walking around their house on a scale from 0 to 100% [32]. Both BBS-14 and ACB-1 are individual items extracted from validated measurement scales. The following equation for the prediction of fall risk was used [30]:(1)$$Likelihood\;of\;Falling\;\left(\%\right)=\frac{\exp\left(-7.519+0.026\left(simple\;RT\right)-0.071\left(ABC\text{-}1\right)-2.139\left(BBS\text{-}14\right)\right)}{1+\exp(-7.519+0.026\left(simple\;RT\right)-0.071\left(ABC\text{-}1\right)-2.139\left(BBS\text{-}14\right))\;}\times 100$$

#### 2.3.2. Arm and Hand Function

The 9-HPT is a quantitative measure of upper extremity (arm and hand) function [33]. Both the self-determined dominant and non-dominant hands were tested twice in an alternating order starting with the dominant hand. To complete the 9-HPT, the subjects were seated in front of the test board with nine empty holes and a small container holding nine pegs. On a start command, the subjects began to pick up each peg one at a time and place them into the holes as quickly as possible. After all nine pegs were placed into holes, the subjects quickly removed the pegs one at a time and placed them back in the container. The total time to complete the entire task was recorded.

#### 2.3.3. Cognitive Function

Cognition was evaluated using the Deary–Liewald simple and choice RT task [29] and the Flanker Inhibitory Control and Attention Test [34]. For simple RT, a white square was positioned in the middle of a computer screen. Subjects were required to press the space bar on the computer laptop keyboard (Dell Latitude 7490, Dell, Round Rock, TX, USA) as soon as they saw a black cross appear in the white box. The inter-stimulus interval for the cross appearance randomly ranged between 1 and 3 s. The subjects performed eight familiarization trials before completing the 20 measurement trials. The time from the cross appearance to the pressing of the space bar was recorded by the computer program and averaged over these 20 trials. During choice RT, four white squares were positioned horizontally in the middle of a computer screen. Four keys on a standard computer keyboard corresponded to the different squares: the “z” key corresponded to the square on the far left, the “x” key to the square second from the left, the “comma (,)” key to the square second from the right, and the “full stop (.)” key to the square on the far right. The subjects placed one finger over each key and were asked to respond as quickly and as accurately as possible to a black cross that randomly appeared in one of the four squares by pressing the corresponding key. The inter-stimulus interval again randomly ranged between 1 and 3 s. The computer program recorded the response times and if the response was correct. The subjects experienced eight practice trials before completing 40 measurement trials. The average of these 40 trails represented the choice RT.

The Flanker Inhibitory Control and Attention Test assesses the ability to inhibit visual attention to irrelevant task dimensions. On each of the 50 trials, a central directional target (arrow) was flanked by similar stimuli on the left and right (five arrows total). The task was to indicate the direction of the central arrow by pressing the “A” key (arrow pointing left) or the “L” key (arrow pointing right). The subjects rested a finger of their dominant hand on a centrally located marker and were instructed to return to this marker after responding to each set of stimuli. On congruent trials, the flankers faced the same direction as the target arrow. On incongruent trials, they faced the opposite direction. The average RTs for congruent and incongruent trials were recorded, and the difference between these (i.e., the Flanker Effect) was calculated by the program.

#### 2.3.4. Static Posturography

Static posturography was performed on a balance board (Balance Tracking Systems, San Diego, CA, USA). Subjects stood quietly on the balance board for 60 s with their arms folded and eyes open and looking at a symbol placed centrally at eye height on the wall in front of them. The primary outcomes from this test included the center of pressure path length in the anterior–posterior (AP-Path) and medial–lateral (ML-Path) directions as well as the area of an ellipse that encapsulated 95% of the 2D area explored (COP_area_).

#### 2.3.5. Gait

Gait was assessed via a 30-m walk test. During this test, subjects were instructed to walk 30 m using their normal, comfortable walking speed while wearing six inertial sensors (OPAL inertial motion units; APDM, Portland, OR, USA) that were used to assess gait speed and stride length. One sensor was placed on the chest/sternum, one on the lower back (5th lumbar vertebra), one on each wrist, and one on top of each foot. In anticipation that some subjects may have existing medical conditions that would make multiple walking trials infeasible (e.g., pain during gait in the Users), only one walk was performed for all subjects.

### 2.4. Statistical Analysis

All outcomes were tested and visually assessed for normality with the Shapiro–Wilk test and Q-Q plots *a priori*. The normality assessment revealed that BBS-14, Flanker Effect, fall risk, and COP_area_ variables did not meet the normality assumption. Therefore, these variables were tested with the nonparametric Mann–Whitney *U* test, with a common language effect size indicator (*A*). The *A* effect size indicator represents the probability that a random data point from one group will be larger than a random data point from another group and is an appropriate effect size for nonparametric analyses [35,36]. The value of *A* ranges between 0.0 and 1.0, with 0.5 interpreted as no effect (i.e., 50% probability) and either extreme (0.0 or 1.0) interpreted as complete separation of the groups. The remaining normally distributed variables were analyzed with independent *t*-tests, accompanied by Cohen’s **d** as an effect size (**d** < 0.2 = small, 0.5 = medium, > 0.8 = large). All statistical tests compared differences between Users and Non-Users, significance for this exploratory analysis was accepted at *p* < 0.05, and analyses were performed with GraphPad Prism 8.1.2 (GraphPad Software, San Diego, CA, USA).

## 3. Results

All of the subjects were US citizens living in Iowa and cannabis consumption was in the form of capsules and/or tinctures. Subjects completed all testing as described. One Non-User subject was unable to complete the 9-HPT with the right side because of a recent shoulder surgery. Choice RT for one User subject was not obtained because this subject experienced a substantial disease-related tremor (essential tremor) and could not maintain the required finger positions. Flanker data for one Non-User subject were removed as outliers because the Flanker Effect variable was negative (i.e., faster reaction time in incompatible vs. compatible trials). The results indicated that Users had a higher fall risk than Non-Users (*p* = 0.005, *A* = 0.91), that Users had poorer balance performance on BBS-14 than Non-Users (*p* = 0.008, *A* = 0.89), and that Users walked with a slower velocity than Non-Users (*p* = 0.03, **d** = 1.2; Figure 1). The effect sizes are interpreted as a 91% probability of Users having a higher likelihood of falling than Non-Users (*A* = 0.91), an 89% probability of Users exhibiting poorer performance on BBS-14 than Non-Users (*A* = 0.89), and a large difference in gait velocity (**d** = 1.2). Table 2 displays the central tendency (mean or median), variability (SD or range), significance, and effect size (**d** or *A*) for all of the tested variables.

## 4. Discussion

This study compared fall risk and cognitive and motor functions of older (>50 years) chronic Users with older Non-Users. The novel observations were that Users had a higher fall risk than Non-Users, using the model developed by Lajoie and Gallagher [30], and that Users exhibited slower gait velocity during the 30-m walk test than Non-Users. However, no significant differences in cognitive function were found.

It is likely that cannabis use contributed to the between-group difference in balance and gait velocity, and several physiological mechanisms support this assumption. For example, endocannabinoids are correlated with baseline activity levels of the spinal locomotor circuitry, with CB1 receptor activation and antagonism modulating increased and decreased locomotor frequency, respectively [37]. CB1 receptors are highly prevalent in movement-related brain regions [26] and in the dorsal and ventral horns of the spinal cord [27]. Thus, consumption of cannabis may result in alterations of the normal rhythmic neural activity of the locomotor circuitry. Cannabinoids can also depress motoneurons by modulating glycinergic [37] and glutamatergic [38] signaling via alterations in CB1 receptor activity. Additionally, CB1 receptor activation might also influence descending signals from the motor cortex, via the modulation of neurotransmitter release in basal ganglia and motor cortex neurons [39,40]. Additionally, it is known that cannabis intoxication results in acute motor deficits, including changes in balance [41]. Indeed, an acute THC concentration-dependent disturbance in balance has been observed, with increasing levels resulting in more body sway [42,43]. This might stem from the activation of CB1 receptors in movement-related brain regions by this intoxicating cannabis compound. Chronic cannabis use in humans is also associated with decreased white matter density in the left parietal lobe [44] and long-term changes in cognitive function (e.g., memory and executive function) [45]. Thus, it is conceivable that functional changes in the motor system might occur. However, the long-lasting disturbances in functional movement tasks, such as balance and gait, in older populations is currently unknown. This is the first study that observed an increased fall risk, poorer balance control, and slower gait velocity in older chronic cannabis users, suggesting that cannabis may, indeed, have long-term effects on both gait and balance in this population. It remains to be seen if these deficits in motor function result in an increase in fall incidence and fall-related injury.

Falls have previously been associated with (1) psychological distress (depression, anxiety, and concern about falling) [46,47], (2) changes in cognitive function (especially executive function [48]), and (3) reduced balance control and gait [49]. A marked deficit in just one of these domains is sufficient to increase fall risk, and concurrent deficiencies in sensorimotor, cognitive, and affective functions are common in older people and contribute to falls [50]. Furthermore, multiple mild–moderate severity deficits can also cumulatively increase fall risk [51].

Static postural control relies upon visual, sensory, and vestibular system inputs [52,53]. With aging, the performance of the sensory system declines, and postural instability and gait limitations become more prevalent. However, the results of this study suggest that chronic cannabis use further exacerbates poor balance control found in normal aging. Specifically, this study found a higher fall risk in chronic cannabis users, based on a validated fall risk model [30]. This result was likely driven by the differences in one leg stance between the groups (Figure 1, Table 2). This impaired ability is concerning given that poor performance on one leg standing tests has previously been associated with injurious falls [54].

Furthermore, Users in this study walked slower during the 30-m walk test. Gait has been used as a marker for physical function and a predictor for falls and even mortality [55,56]. Gait is a highly complex activity that is influenced by changes in central and peripheral nervous systems, the skeletal muscle system, and the brain (e.g., the basal ganglia or motor cortical regions) [55,57,58]. Gait speed/velocity is one of the most studied variables in aging research [55,59]. Indeed, reductions in self-selected gait speed, driven by shortened stride length [60] and altered step width [61], and fast gait [61,62,63], which is relevant to daily life (e.g., a velocity > 1.2 m/s is required to safely traverse lighted intersections [64,65]), have been reported. Thus, the difference in gait speed between the Users (0.96 ± 0.30) and Non-Users (1.26 ± 0.17) in this study is likely relevant to their increased fall risk [56] (Figure 1, Table 2).

Studies in younger people have shown that cannabis impairs cognition, likely due to the detrimental effects of Δ9-tetrahydrocannabinol (THC), the main psychoactive compound in cannabis [66,67]. Acute cannabis intoxication is associated with greater impulsivity and poorer working memory in young chronic cannabis users [68], and non-acute effects have been shown to persist in this population [66,69]. In addition, a recent review [70] also found impaired verbal memory and brain activity alterations in subjects ≤ ~37 years old. However, comparatively little is known about the effects of cannabis on cognition in older adults (≥50 years). In this study, cognitive function, as assessed with RT and Flanker tests, was not different between Users and Non-Users. A few possible explanations for this finding are: (1) All cannabis products used by the subjects included at least some amount of cannabidiol (CBD), which has anti-inflammatory properties [71] and may diminish the negative cognitive influences of low-grade inflammation seen in aging [72,73]. (2) Aging is accompanied by alterations in multiple components of the dopamine system [74,75]. In addition, chronic cannabis use has been correlated with a blunting of the dopamine system [76] and a decline in dopamine has been associated with poorer cognitive and motor performance [77,78]. However, these studies included young and middle-aged short-term users, and whether the blunted dopamine release and synthesis persist in older chronic users remains unknown. However, the results of our study agree with other findings. Thayer et al. [79] found no significant differences between older cannabis users and older non-users on attention, episodic memory, working memory, vocabulary, reading, executive function, and processing speed. Similarly, Burggren et al. [80] also found equivalent results between older adults with a history of heavy cannabis use and older controls on encoding and delayed memory, processing speed, and executive function. Furthermore, the mini-mental state examination (MMSE) scores of the cannabis users did not differ from controls [80]. (3) The results of the Flanker Effect (*p* = 0.07, *A* = 0.71) also suggest that the cognitive tests may have been underpowered. Further research examining the interaction between chronic cannabis use and advanced age on cognition is clearly needed.

The most prominent limitation of the current study is the relatively small sample size, which necessitates caution when interpreting the results. A larger sample would allow for the detection of smaller changes and greater confidence in the differences found in the balance and gait variables and lack of group differences in cognitive performance. In addition, although most subjects (7/8) in this study reported using cannabis for pain relief, their different physical impairments and the variety of THC/CBD ratios of the cannabis products used may have resulted in a heterogenous Users group. Specifically, although many subjects used THC-dominant products, it is possible that the increased fall risk and impaired gait parameters found in this study are not attributable to cannabis use but are instead the result of an accelerated aging process and/or greater physical impairments. Moreover, poly-drug use (e.g., history of cannabis and hallucinogens) and uncertainty about the dose (e.g., in mg) and potency of cannabis used are limitations that affect all studies on human cannabis users. The biological sex of the subject may contribute to their cannabis use and response [81]. However, this study included people from both sexes and how this variable might have influenced the findings is unclear. Additionally, whether the Users have discontinued use since participation is unclear, although the relatively long duration of use for medical purposes (i.e., >6 months, as per the inclusion criterion and >10 years average usage time, Table 1) suggests cessation is unlikely. Lastly, our study included subjects that may more appropriately be described as middle-aged (mean age = ~60 years), and these results may not generalize to the oldest of older adults. Given the normative age-related decline in motor and cognitive functions, it is logical to speculate that cannabis-induced changes would be even greater in an older cohort. Still, the increased fall risk found in the Users, despite their age and comorbidities, is concerning and requires further study.

Future, preferably longitudinal, studies investigating the long-term effects of cannabis with different THC/CBD ratios on cognitive and motor functions in older adults are needed. Ultimately, given the recent increase in cannabis use among older adults, future human research should examine the association between both early and later-life cannabis use, with more homogenous user and non-user groups. This research should employ cognitive and motor assessments, coupled with neuroimaging or other mechanistic tools to assess associations between altered brain function and behavior [82], to develop a more comprehensive understanding of the effects of cannabis use in older adults. It is also of interest to examine whether declines in balance and gait observed in the Users can be minimized with targeted rehabilitation. Future research should also account for confounding factors, including acute vs. non-acute effects, cognitive reserve, route of administration, and THD/CBD ratio preference.

## 5. Conclusions

In summary, this study found that, compared to older non-users, older adults who chronically use cannabis have increased fall risk and slower gait velocity, but without differences in cognitive function. These effects are purported to be additive to the poorer balance control and slower gait velocity associated with normal aging. Future mechanistic (e.g., neuroimaging) investigations on the short- and long-term effects of the use of a variety of cannabis products (e.g., THC/CBD ratios and routes of administration) on cognitive and motor symptoms, as well as fall incidence, are required to inform older adults, and their clinicians and caregivers, about the potential risks of cannabis use and to help develop guidelines for cannabis use for medical purposes.

## Figures and Tables

**Figure 1 brainsci-11-00134-f001:**
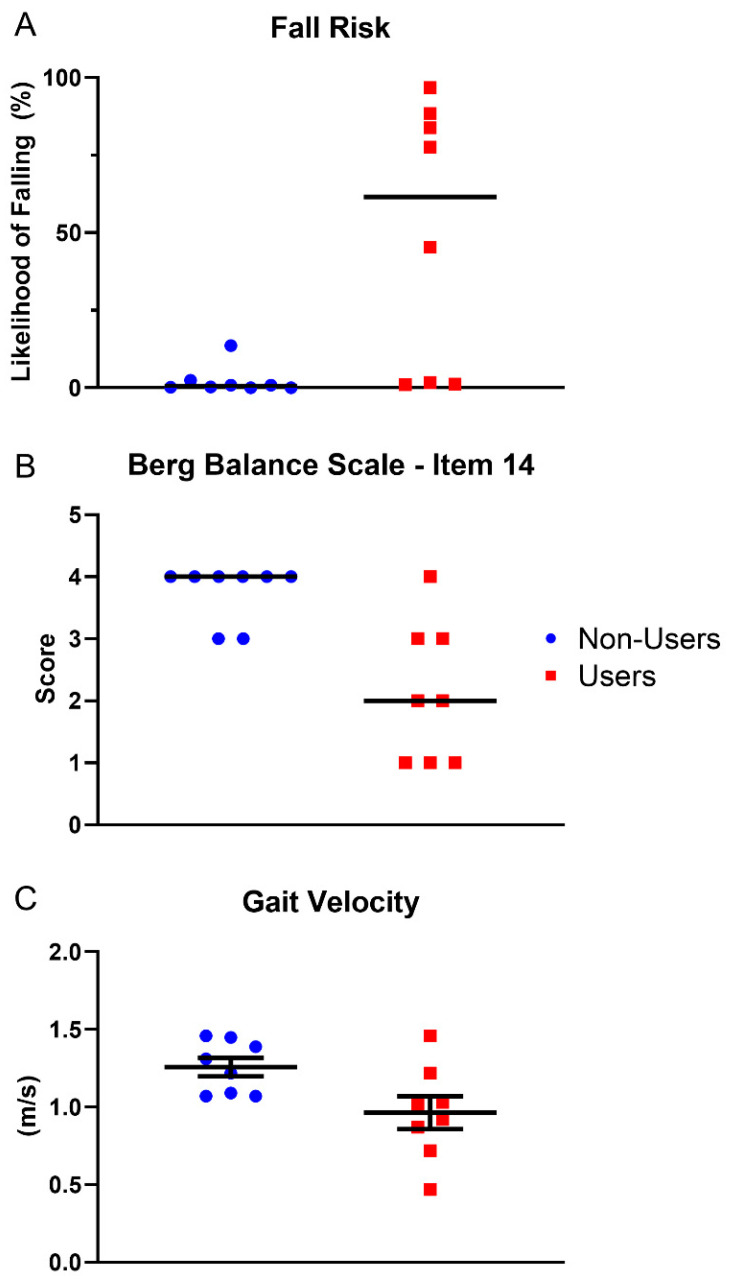
Plots of significant differences between cannabis Users vs. Non-Users in (**A**) fall risk, (**B**) Berg Balance Scale—Item 14, and (**C**) gait velocity. Individual subject scores within each group are represented by the different data points (n = 8 per group). The horizontal bars in A and B indicate the median and the bars in C represent mean ± SEM.

**Table 1 brainsci-11-00134-t001:** Subject demographic information stratified by user status. Data are mean ± SD.

Demographic	Non-Users	Users
Sex (M/F)	3/5	3/5
Age (years)	60.5 ± 4.7	59.6 ± 4.8
Height (cm)	168.9 ± 10.6	169.5 ± 11.4
Weight (kg)	82.5 ± 20.7	92.8 ± 24.9
Duration of cannabis use	n/a	10.40 ± 12.6
Uses per week (days)	n/a	4.9 ± 2.5
Uses per day (times)	n/a	1.4 ± 0.7
THC dominant (n)	n/a	4
THC = CBD (n)	n/a	2
CBD dominant (n)	n/a	1
Multiple types (n)	n/a	1
Medical reasons for use (n)	n/a	Pain (7), PD (1)

THC = Δ-9-Tetrahydrocannabinol, CBD = cannabidiol, PD = Parkinson’s disease.

**Table 2 brainsci-11-00134-t002:** Central tendency, variability, significance, and effect size (**d** or *A*) for all variables compared between cannabis Users and Non-Users. Data are mean ± SD or median (range). Significant *p*-values are in bold.

Variable Name	Users	Non-Users	*p*-Value	Effect Size
ABC-1 (%) *	83.3 ± 15.4	85.6 ± 14.5	0.76	**d** = 0.2
BBS-14 (score) *	2 (1–4)	4 (3–4)	**0.008**	*A* = 0.89
Simple RT (ms) *	629.4 ± 38.7	668.1 ± 65.0	0.17	**d** = 0.7
Choice RT (ms)	628.3 ± 204.4	623.9 ± 88.4	0.96	**d** = 0.0
Likelihood of Falling (%)	61.5 (1.1–96.8)	0.5 ± (0.05–13.6)	**0.005**	*A* =0.91
Flanker-Compatible (ms)	959.3 ± 189.3	963.0 ± 63.8	0.96	**d** = 0.0
Flanker-Incompatible (ms)	1100.6 ± 224.5	1031.1 ± 59.0	0.44	**d** = 0.4
Flanker Effect (ms)	140 (41–333)	72 (2–112)	0.07	*A* = 0.71
9-HPT, D (s)	24.4 ± 4.5	22.0 ± 4.0	0.30	**d** = 0.6
9-HPT, ND (s)	26.4 ± 4.1	23.2 ± 3.7	0.13	**d** = 0.8
Gait Velocity (m/s)	0.96 ± 0.30	1.26 ± 0.17	**0.03**	**d** = 1.2
Stride Length (m)	1.1 ± 0.2	1.3 ± 0.2	0.06	**d** = 1.01
AP-Pathlength (cm)	2.5 ± 0.8	2.2 ± 0.6	0.47	**d** = 0.4
ML-Pathlength (cm)	1.1 ± 0.4	0.9 ± 0.3	0.28	**d** = 0.6
COP_area_ (cm^2^)	1.6 (0.9–3.8)	1.1 (0.4–6.6)	0.38	A = 0.36

ABC-1 = Activities Balance Confidence scale, question 1; BBS-14 = Berg Balance Scale, Item 14; RT = reaction time; 9-HPT = 9-Hole Peg Test; D = dominant hand; ND = non-dominant hand; AP = anterior–posterior; ML = medio–lateral; COP_area_ = area of an ellipse that encapsulates 95% of the 2D center of the pressure trace. Variables marked with an asterisk (*) were used in the Likelihood of Falling calculation [28].

## Data Availability

Data for this study will be made available upon request by contacting the corresponding author.

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
