# Peer review of "Increased Likelihood of Falling in Older Cannabis Users vs. Non-Users"

_brainsci, 2021, doi:10.3390/brainsci11020134_

Round 1

Reviewer 1 Report

Abstract:

The null results (cognitive tasks) should be addressed in the abstract.

Methods:

Were the participants screened for other drug use, including prescription medicines and alcohol consumption.

I am a little unclear as to whether the group differences could be attribute to cannabis use and/or other variables associated with a clinical diagnosis (Pain/PD). Could there be other exogenous variables that are driving differences based upon one group having no clinical issues. A possible control for this would be a third group who were matched for pain and PD? This at least needs to be clarified and addressed.

Apart from the small n this is a well designed and important contribution to the literature. Both cannabis and age related cannabis use needs to be explored at this work points to the importance of this line of research despite the limitations of a small n and possible confounding variables.

Author Response

We thank the reviewer for the helpful comments and suggestions which helped to improve the quality of this paper. Please see below our responses. Changes in the manuscript are highlighted in red.

Abstract:

The null results (cognitive tasks) should be addressed in the abstract.

We added “No significant differences in cognitive function were found” to the abstract (line 20).

Methods:

Were the participants screened for other drug use, including prescription medicines and alcohol consumption.

Participants who used other drugs and/or consumed alcohol were excluded. This was an exclusion criterion and has been added (line 96).

I am a little unclear as to whether the group differences could be attribute to cannabis use and/or other variables associated with a clinical diagnosis (Pain/PD). Could there be other exogenous variables that are driving differences based upon one group having no clinical issues. A possible control for this would be a third group who were matched for pain and PD? This at least needs to be clarified and addressed.

We agree that factors other than cannabis use might, at least in part, explain some of our findings. We have already mentioned this as a study limitation (lines 306 – 309)

Reviewer 2 Report

Dear Authors, 

The research article titled "Increased Likelihood of Falling in Older Cannabis Users vs. Non-Users" is well written and sounds good.

However, as you clearly stated in the discussion and conclusion sections, there are some limitations.

Furthermore, I have some suggestions and questions as well concerning your paper.

1- References needed concerning the cannabinoids effects on: the memory alterations (doi: 10.3390/brainsci10020102); sex and gender differences in response to cannabinoids (doi: 10.3390/brainsci10090606); and how cannabis can affect brain and behavior as reported by Cozzolino and colleagues (doi: 10.3390/brainsci10110834).

2- Considering the inclusion criteria, in the methods section, the Authors stated that the patients have to be able to speak, read, and understand English. On this regard, should be useful to add the nationality of patients. 

3- The asterisks in the table what do they mean? Please add the significance in the appropriate legend section.

4- Could you better explain how cannabis was administered (smoke, oil, others?)

5- It should be fine to add if the patients/participants have discontinued the therapy.

6- Have the Authors genetically analyzed the patience, since it has been reported that some cannabinoid receptors polymorphisms can influence its efficacy (doi: 10.6323/JCRP.2014.1.3.01).

Author Response

We thank the reviewer for the helpful comments and suggestions which helped to improve the quality of this paper. Please see below our responses. Changes in the manuscript are highlighted in red.

  • References needed concerning the cannabinoids effects on: the memory alterations (Blest-Hopley et al. 2020)); sex and gender differences in response to cannabinoids (Fattore et al. 2020); and how cannabis can affect brain and behavior as reported by Cozzolino and colleagues (Collizzi 2020).

Text and the citations were added to the discussion section (lines 279 – 281; lines 315 – 317; lines 332-333).

  • Considering the inclusion criteria, in the methods section, the Authors stated that the patients have to be able to speak, read, and understand English. On this regard, should be useful to add the nationality of patients. 

This information has been added (line 191).

  • The asterisks in the table what do they mean? Please add the significance in the appropriate legend section.

These asterisks indicate that these variables were used in the Likelihood of Falling calculation, as previously indicated in the table footer (lines 215 – 216). In addition, this table also contains a column of the p-values. However, we have now drawn attention to the significant values with bold text.

  • Could you better explain how cannabis was administered (smoke, oil, others?)

This information was added at line 191 – 192.

  • It should be fine to add if the patients/participants have discontinued the therapy.

This was a cross-sectional observational study that did not have a follow-up in which we might have determined discontinuation. We have added a statement about the lack of discontinuation information to the limitations (lines 317 – 320).

6- Have the Authors genetically analyzed the patience, since it has been reported that some cannabinoid receptors polymorphisms can influence its efficacy (doi: 10.6323/JCRP.2014.1.3.01).

Genetic analysis of our subjects was outside of the scope of this study. In addition, the reference provided discusses genetic polymorphisms in susceptibility to opioids, which does not directly inform cannabinoid receptor polymorphisms.

Round 2

Reviewer 1 Report

Thanks so much for working with me on the revisions!

Reviewer 2 Report

Dear Authors, 

The manuscript has been improved significantly.